# Application of Analytic Hierarchy Process to Rank Fire Safety Factors for Assessing the Fire Probabilistic Risk in School for the Blind Building: A Case Study in Thailand

**Arroon Ketsakorn** [1,*] **and Rujipun Phangchandha** [2]

1   Faculty of Public Health, Thammasat University Rangsit Center, Klong Luang, Pathumthani 12121, Thailand
2   Department of Safety Technology and Occupational Health, Faculty of Industrial Technology,
    Suan Sunandha Rajabhat University, Uthong Nok Rd., Dusit, Bangkok 10300, Thailand; rujipun.ph@ssru.ac.th
*   Correspondence: arroon.k@fph.tu.ac.th; Tel.: +66-2564-4440

**Abstract:** Fires are the leading cause of death, serious injury, and property damage. In the past, schools, temples, and government offices had more frequent fires than they should. Statistics showed that the number of fires between 2017 to 2022 amounted to 13,593 cases which mostly occurred in schools, temples, and government offices (40.0% of all buildings). Moreover, it causes more damage among the blind, who have limited vision. Therefore, the cross-sectional purpose of this study was to assess the fire risk in school for the blind. The fire checklists, brainstorming, and analytic hierarchy process (AHP) were applied to estimate the fire risk in the school for the blind building. The findings revealed an inherent fire hazard factor (fire probabilistic risk scores = 3.2830) and evacuation factor (fire probabilistic risk scores = 3.3178) that were acceptable risks, except the fire control factor (fire probabilistic risk scores = 1.4320) was an unacceptable risk (score of less than 2.5). The unacceptable risk may cause impacts to life, health, property, and public communities. Eventually, efforts should be made to supervise those risk factors by designing suitable activities to reduce undesirable conditions in schools for the blind.

**Keywords:** fire probabilistic risk; fire safety factor; fire checklist; analytic hierarchy process

## 1. Introduction

Fires in Thailand tend to happen more frequently and severely mainly because of unsafe acts and unsafe conditions. For a megacity with a high permanent and temporary population like Bangkok, fire management becomes more crucial. Fires may cause many injuries and casualties as well as damage to property and assets, in addition to harm to the overall economic system. The Department of Disaster Prevention and Mitigation, Ministry of Interior of Thailand [1], as the administrative body responsible for the city, undertakes the mission to protect the population from public disasters and make life safer. Reports from this government organization indicated that the number of fires between 2017 to 2022 amounted to 13,593 cases which mainly occurred in schools, temples, and government offices (40% of all buildings). Moreover, it causes more damage among the blind, who have limited vision.

Previous studies in the fire probabilistic risk assessment sector have predicted the risk status in the buildings where the occupants are to able complete the fire evacuation when fire occurs. Both deterministic and stochastic fire risk assessment approaches were used to determine the risk status without brainstorming and evaluating the consistency of fire experts' judgement, which causes the fire risk status to be inaccurate, such as Prashant Tharmarajan's [2] study on the essential aspects of fire safety management in high-rise buildings; this study used the checklist technique to survey the level of fire experts' opinions for determination of fire safety management. However, the consistency of the opinion level was not assessed to reduce the bias in the fire experts' judgement. John Moore [3]

conducted a study on the assessment of fire safety and evacuation assessment in nursing homes; this paper used a building survey to establish fire safety facilities, and analysis of the fire-related documentation, interviews with staff to establish existing fire safety procedures, and fire risk assessment were carried out in this study. Just one aspect that this study did not address is that the acquisition of fire experts' opinion levels was undertaken to reduce bias in the assessment of individual fire expert ratings. Yeung Cho Hung [4] studied fire safety management of public rental housing in Hong Kong. Fault tree analysis (FTA) was used to study the effect of fire safety management on the reduction in fire risk level. FTA is a method of identifying the possible cause of a system failure. By using FTA, the assessing team can determine what factors contributed to an event (known as a failure or top event), and the probability of it occurring. However, the various probabilities were estimated to evaluate the risk level without considering the consistency assessment to reduce the bias of individual fire experts' judgement. Tanima Abdul Wahed [5] focused on the impact of facility management on fire safety crisis in the industry. A qualitative questionnaire survey was conducted to obtain the real picture of fire management practices in Bangladesh, but there were no reference case studies demonstrating an application of the fire probabilistic risk assessment. There is also a study by Ayyappa Thejus Mohan [6] that focused on risk acceptance in fire safety engineering. Traditional deterministic fire engineering was used as a tool for data collection from the experience of the fire safety profession as well as through a continuous process of trial and error for assessing the fire risks. However, the likelihood of harm for assessing the risk was not assessed for consistency in individual scoring.

All the above research focuses on the fire probabilistic risk assessment and management in different buildings but there is no focus on a new tool used for calculating the level of risk. It obviously can be seen that there is still a lack of reliable fire safety assessment tool for reduction of bias in individual fire experts' judgement. This is an important consideration to ensure the safety of occupants in buildings, especially blind people who have limited vision. Assessing the fire risk is important in the occupational health and safety field. Therefore, this study aimed to assess fire risk in a school for the blind building, including applying a fire probabilistic risk assessment technique using the checklist method, which is a hazard identification technique used to reduce failure by compensating for potential limits of human memory and attention. It helps to ensure consistency and completeness in carrying out a task. This technique examines fire hazards according to laws and standards and is therefore appropriate and consistent with the task of applying it for risk assessment as compared to other techniques. However, the checklist has limitations in determining the probability of the event, where, sometimes, some fire safety factors are unable to determine these values. Therefore, it is necessary to apply other techniques to determine the probability of events.

The analytic hierarchy process (AHP) proposed by Thomas Saaty [7] is a versatile tool for dealing with complex decision-making problems. It is a hierarchical structure of goals, objectives, main criteria, sub-criteria, and alternatives. The AHP assists deciders to look for the best matches for their goal by eliminating biased decisions. It provides an extensive and methodical structure for construction of a decision issue, for illustrating and estimating its elements, for implicating those elements to overall goals, and for assessing alternative solutions. The integration of AHP and expert analysis is frequently found in several studies, especially in the fields of waste management [8,9], environmental deterioration [10], water resource management [11–13], project management selection [14,15], and disaster response [16], while the applications in the domain of determination of fire safety factors for assessing the fire risk, fire probabilistic risk, are less consolidated. This study eventually applies an AHP to determine the relative weight of each fire hazard factor from a checklist and determine the probabilistic risk of fire. It also proposes measures for managing the fire risk by designing suitable activities to reduce undesirable conditions in a school for the blind. One of the advantages of AHP application is its stability and flexibility regarding changes within, and addition to, the hierarchy of structure. Another advantage of AHP is its ability to rank main factors in accordance with the needs of policymakers, leading to

more precise decision making. However, AHP may have some weak points. One of them is the complexity of the method, which may be inconvenient in implementation. When more than one person is working with this method, it can be complicated by different opinions on the weighting of each of the criteria.

## 2. Materials and Methods

### 2.1. Study Design and Setting

A cross-sectional study was conducted between June and December 2022 in a school for the blind building in Bangkok, Thailand. The study area was a classroom of a blind school building according to the Building Act, B.E. 2522 of Thailand. This classroom is ventilated with natural air. There are three stories, 79.5 m long, 16 m wide, and 3 m high, the first and third floor have three fire escape stair exits, the second floor has two fire escape stair exits, and one of the fire escape stair exits was blocked off to create a new classroom on the second floor, as shown in Figure 1. In this study, the classroom was selected by considering the prioritization of fire safety problems, namely, attention to problems of staff, attention of management to problem solving, and problem size.

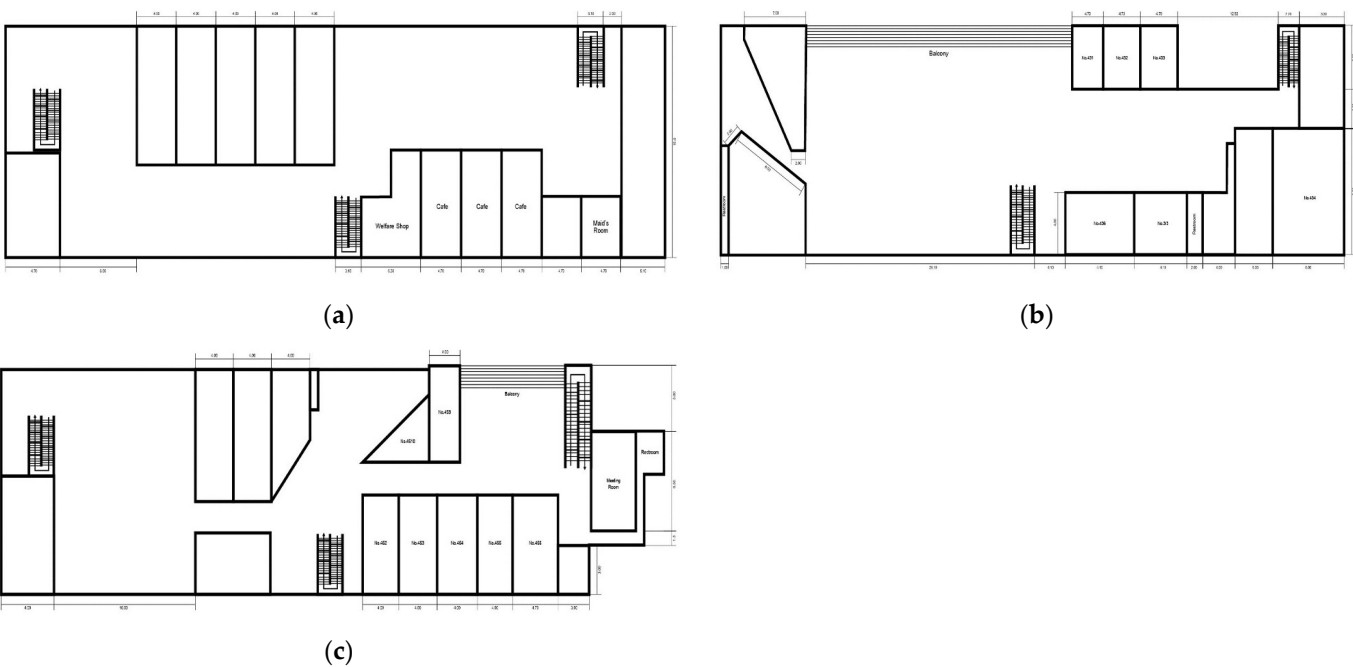

**Figure 1.** School for the blind building floor plan: (**a**) the first-floor plan; (**b**) the second-floor plan; (**c**) the third-floor plan.

### 2.2. Instrument Development

An instrument for data collection consisted of a checklist and using Expert Choice Version 11 in computing the relative weight from an AHP structure and determining the risk of fire in the school for the blind building.

Expert Choice is software for cooperative decision-making solutions that are based on multicriteria decision making. This software helps organizations make better decisions and manage risk with speed, transparency, and alignment. Expert Choice implements the AHP [17] and has been used in fields such as manufacturing [18], environmental management [19,20], shipbuilding [21] and agriculture [22]. This collaboration software was created by Thomas Saaty and Ernest Forman in 1983 [23] and supplied by Expert Choice Corporation. This research used this tool to compute the relative weight by brainstorming from the checklist. The AHP provided a structural framework, as shown in Figure 2, for setting priorities on each level of the hierarchy using pairwise comparisons that were quantified using the scale of 1–9 in Table 1. The pairwise comparisons between the decision

criteria can be conducted by asking fire safety experts questions. The answers to those questions are given in an m × m pairwise comparison matrix, stated as follows in Equation (1).

$$A = \left(a_{ij}\right)_{m \times m} = \begin{array}{c} C_1 \\ C_2 \\ \vdots \\ C_m \end{array} \begin{bmatrix} a_{11} & a_{12} & \dots & a_{1m} \\ a_{21} & a_{22} & \dots & a_{2m} \\ \vdots & \vdots & \dots & \vdots \\ a_{m1} & a_{m2} & \dots & a_{mn} \end{bmatrix} \qquad (1)$$

where $a_{ij}$ denotes a numerical judgment on $w_i/w_j$ with $a_{ii} = 1$ and $a_{ij} = 1/a_{ji}$ for $i, j = 1, \dots, m$. On condition that the pairwise comparison matrix $A = (a_{ij})_{m \times m}$ fulfils $a_{ij} = a_{ik} a_{kj}$ for any $i, j, k = 1, \dots, m$, afterwards, $A$ is said to be magnificently consistent; otherwise, it is said to be inconsistent. The pairwise comparison matrix $A$ and the weight vector $W$ can be computed by solving Equation (2):

$$AW = \lambda_{\max} W \qquad (2)$$

where $\lambda_{\max}$ is the maximum eigenvalue of $A$. The answers from experts may not be competent enough to contribute to ideally accordant pairwise comparisons, so it was dictated that the pairwise comparison matrix $A$ must have an acceptable consistency, which can be examined by the following consistency ratio (*C.R.*), as expressed in Equation (3).

$$C.R. = \frac{(\lambda_{\max} - n)/(n-1)}{RI} \qquad (3)$$

where *RI* is a random inconsistency index; those values were deviated with the order of the pairwise comparison matrix. The RI values for the pairwise comparison matrices with the order from 1 to 10 are shown in Table 2. Whether or not *C.R.* ≤ 0.1, the pairwise comparison matrix is considered to have an acceptable consistency; on the other hand, it must be corrected.

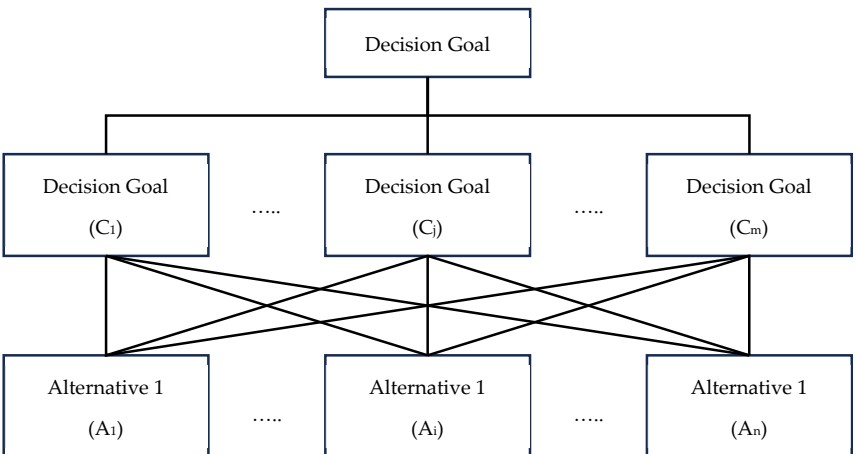

**Figure 2.** A three-level decision hierarchy.

**Table 1.** Numerical value pairwise comparison scale [7].

| Numerical Rating | Description for Risk Factor Evaluation |
| --- | --- |
| 1 | Equivalently |
| 2 | Equivalently to neutrally more |
| 3 | Neutrally more |
| 4 | Neutrally to greatly more |
| 5 | Greatly more |
| 6 | Greatly to very greatly more |
| 7 | Very greatly more |
| 8 | Very greatly to extremely more |
| 9 | Extremely more |

**Table 2.** Random inconsistency index for pairwise comparison matrices [7].

| Order (*n*) | Random Inconsistency Index (*RI*) |
|---|---|
| 1 | 0 |
| 2 | 0 |
| 3 | 0.58 |
| 4 | 0.90 |
| 5 | 1.12 |
| 6 | 1.24 |
| 7 | 1.32 |
| 8 | 1.41 |
| 9 | 1.45 |
| 10 | 1.49 |

Decision alternatives can be considered pairwise with regard to each decision criterion. After the weights of decision criteria and the weights of decision alternatives are achieved by using pairwise comparison matrices, the overall weight of each decision alternative with regard to the decision goal can be produced by the following addition weighting method [7], as expressed in Equation (4).

$$w_{Ai} = \sum_{j=1}^{m} w_{ij} w_j, \ i = 1, \ldots, n \tag{4}$$

where $w_j$ ($j = 1, \ldots, m$) are the weights of decision criteria, $w_{ij}$ ($i = 1, \ldots, n$) are the weights of decision alternatives with regard to criterion $j$, and $w_{Ai}$ ($i = 1, \ldots, n$) are the overall weights of decision alternatives. The best decision alternative is the one that has the greatest overall weight relative to the decision goal.

Expert Choice implemented the AHP that was used for computing the relative weight, instead of calculating by hand according to Equations (1)–(4).

*2.3. Order of Operations Steps*

There were three different stages of the research process. The purpose of the first stage was to identify the fire risk using the fire checklist. This contextual information was then used in the second stage of the research for fire probabilistic risk assessment using analytic hierarchy process and brainstorming for obtaining the weight of each factors. In the last stage, the results were finally compiled as fire risk scores to make a decision for conducting fire management in order to reduce the fire risk in the school for the blind building. An order of operation steps is shown in Figure 3.

First stage: We explored the primary and secondary data to design the fire checklist. We explored the primary and secondary data, including reviewing the legal and other requirements of fire, to design the fire checklist. This fire checklist was then given to the fire experts. The fire checklist was used to collect data for identifying the hazards in the school for the blind building.

Second stage: Fire probabilistic risk assessment. We quantified the risk of the common tasks with consideration of probability and relative weight of fire safety factors. Fire probabilistic risk assessment used the brainstorming and analytic hierarchy process to obtain the weight of each factors in the fire checklist.

Third stage: We organized fire probabilistic risk scores into a decision to conduct the fire mitigation. The results from the stage of quantification of fire probabilistic risk assessment were compiled into the fire risk scores to compare with risk standards. Finally, unacceptable risk was calculated within the fire mitigation in order to reduce the fire risk in the school for the blind building.

Fire probabilistic risk assessment in the school for the blind building can be calculated using Equation (5).

$$R = \sum_{i=1}^{n} P_{ci} \times W_{ci} \tag{5}$$

$R$ = Fire probabilistic risk scores.

$n$ = The number of fire probabilistic risk assessment index.

$W_{ci}$ = Weight of fire probabilistic risk assessment indicator $i$, ranging from 0 to 1.

$P_{ci}$ = Scores of fire probabilistic risk assessment indicator $i$, ranging from 1 to 5.

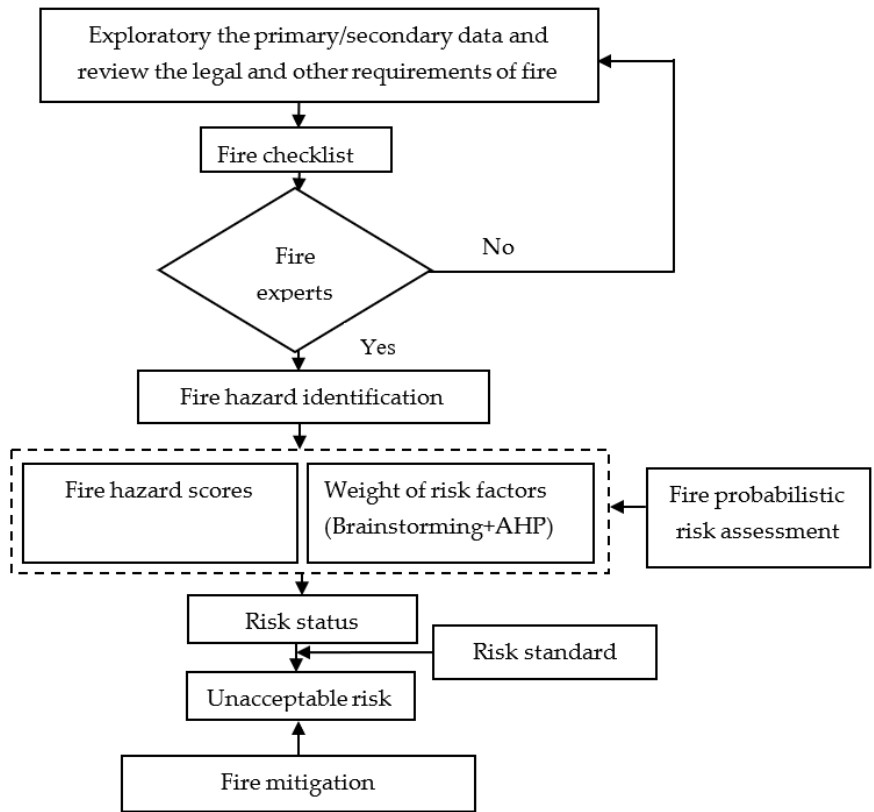

**Figure 3.** Order of operation steps.

The fire risk scores were considered in two elements. Namely, the assessment grade scores were based on brainstorming from seven fire safety experts, meaning that the probability levels were remote possibility (score = 1), possible but not great possibility (score = 2), moderate possibility (score = 3), important possibility (score = 4), and most possible (score = 5), respectively. The scores of fire probabilistic risk assessment are shown in Table 3 and the relative weights of the fire probabilistic risk assessment indicator were obtained from the calculation using Expert Choice V.11. The determination of each weighting criteria and weighting attribute were assumed using a pairwise comparison matrix provided by decision-makers based on the Delphi MAH technique developed by the Rand Corporation Company and the maximize agreement heuristic (MAH). This technique consisted of three steps: the numerical data were acquired from the fire safety experts, all decision data were averaged, and average values from all the decision-makers were considered. The probabilistic risk assessment was performed, and the unacceptable risk level (score less than 2.5) requires further mitigation measures.

**Table 3.** The meaning and score of fire probabilistic risk [18].

| Score | Description |
| --- | --- |
| 1 | Remote possibility |
| 2 | Possible but not great possibility |
| 3 | Moderate possibility |
| 4 | Important possibility |
| 5 | Most possible |

## 3. Results

Assessment of the scores was based on brainstorming from fire safety experts. The fire safety experts consisted of fire safety and building services staff (two persons), firefighters (two persons), fire safety consultants (two persons), and an architect (one person). All experts had more than 5 years' experience in fire safety. Expert Choice V.11 implemented the AHP for computing the relative weight of each of the fire safety factors. The results are shown in Figures 4–6 and Tables 4–8. In addition, there was an example of calculating the relative weight without using a program.

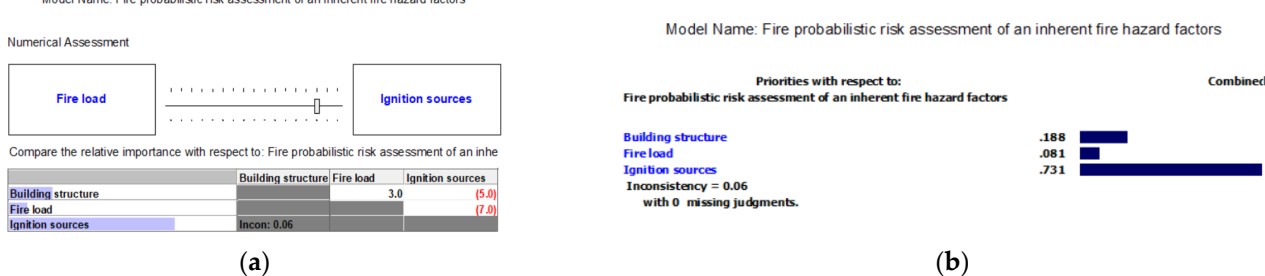

**Figure 4.** Pairwise comparisons of an inherent fire hazard factor: (**a**) pairwise comparisons in Expert Choice V.11 program; (**b**) relative weight calculation using Expert Choice V.11 program.

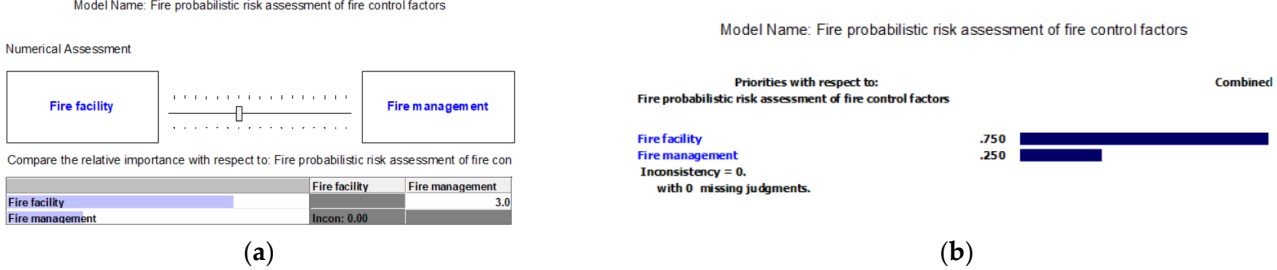

**Figure 5.** Pairwise comparisons of fire control factor: (**a**) pairwise comparisons in Expert Choice V.11 program; (**b**) relative weight calculation using Expert Choice V.11 program.

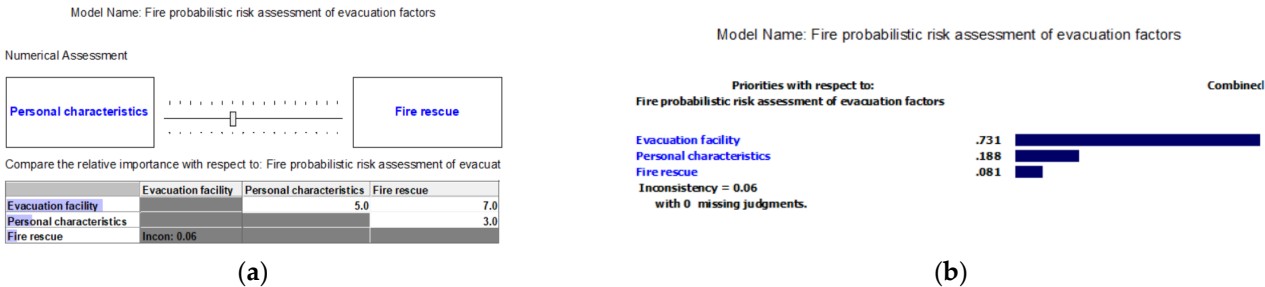

**Figure 6.** Pairwise comparisons of evacuation factor: (**a**) pairwise comparisons in Expert Choice V.11 program; (**b**) relative weight calculation using Expert Choice V.11 program.

The inherent fire hazard factor was categorized into three elements: building structure, fire load, and ignition sources, based on fire safety expert brainstorming. A pairwise comparison matrix for the three decision criteria was provided by decision-makers and was based on the Delphi MAH technique.

The pairwise comparison of the decision criteria showed that the ignition sources were the most important of the decision criteria, with 73.1% significance. The next most important decision criteria were the building structure and fire load, with 18.8% and 8.1% significance.

**Table 4.** Fire probabilistic risk assessment of an inherent fire hazard factor.

| Criteria | Inherent Fire Hazard Factor | Grade Assessment (A) | Weighting Attribute (B) | Scores Attribute (A × B) |
|---|---|---|---|---|
| Building structure | | | | |
| 1 | Height of the blind school building | 4 | 0.0625 | 0.2500 |
| 2 | Multilayer | 4 | 0.1375 | 0.5500 |
| 3 | Fire resistance rating | 3 | 0.3250 | 0.9750 |
| 4 | Hazard classification | 1 | 0.4750 | 0.4750 |
| | Sum of A × B | (C) | | 2.2500 |
| | Relative weight of building structure | (D) | | 0.1880 |
| | Building structure scores | (C) × (D) | | 0.4230 |
| Fire load | | | | |
| 1 | Location area | 3 | 0.1700 | 0.5100 |
| 2 | Interior decoration | 3 | 0.8300 | 2.4900 |
| | Sum of A × B | (C) | | 3.0000 |
| | Relative weight of fire load | (D) | | 0.0810 |
| | Fire load scores | (C) × (D) | | 0.2430 |
| Ignition sources | | | | |
| 1 | Electrical equipment | 3 | 0.6330 | 1.8990 |
| 2 | Type of combustible gas supply | 3 | 0.1070 | 0.3210 |
| 3 | External fire | 5 | 0.6330 | 1.3600 |
| | Sum of A × B | (C) | | 3.5800 |
| | Relative weight of ignition sources | (D) | | 0.7310 |
| | Ignition sources scores | (C) × (D) | | 2.6170 |
| Building structure scores + fire load scores + ignition sources scores | | | | 3.2830 |

**Table 5.** Relative weight scale based on the Delphi MAH technique.

| Criteria (*C.R.* = 0.06 < 0.1) | Building Structure | Fire Load | Ignition Sources | Priority Value |
|---|---|---|---|---|
| Building structure | 1 | 3 | 1/5 | 0.188 |
| Fire load | 1/3 | 1 | 1/7 | 0.081 |
| Ignition sources | 5 | 7 | 1 | 0.731 |
| Sum | 6.33 | 11.00 | 1.34 | 1.000 |

**Table 6.** Maximum eigenvalue ($\lambda_{max}$).

| Calculation Guidelines | Sum of Value | | | Total |
|---|---|---|---|---|
| Vertical sum of value | 6.33 | 11.00 | 1.34 | |
| Horizontal sum of value | 0.188 | 0.081 | 0.731 | |
| Maximum eigenvalue ($\lambda_{max}$) | 1.19 | 0.89 | 0.98 | 3.06 |

**Table 7.** Fire probabilistic risk assessment of fire control factor.

| Criteria | Fire Control Factor | Grade Assessment (A) | Weighting Attribute (B) | Scores Attribute (A × B) |
|---|---|---|---|---|
| Fire facility | | | | |
| 1 | Fire alarm and fire control linkage system | 1 | 0.3520 | 0.3520 |
| 2 | Portable fire extinguisher | 3 | 0.2020 | 0.6060 |
| 3 | Floor plan | 1 | 0.0930 | 0.0930 |
| 4 | Safety sign | 1 | 0.1780 | 0.1780 |
| 5 | Emergency lighting | 1 | 0.1320 | 0.1320 |
| 6 | Lightning protection system | 5 | 0.0430 | 0.2150 |
| | Sum of A × B | (C) | | 1.5760 |
| | Relative weight of fire facility | (D) | | 0.7500 |
| | Fire facility scores | (C) × (D) | | 1.1820 |

**Table 7.** *Cont.*

| Criteria | Fire Control Factor | Grade Assessment (A) | Weighting Attribute (B) | Scores Attribute (A × B) |
|---|---|---|---|---|
| Fire management | | | | |
| 1 | Fire safety inspection | 1 | 0.5400 | 0.5400 |
| 2 | Basic firefighting | 1 | 0.2300 | 0.2300 |
| 3 | Fire drill | 1 | 0.0725 | 0.0725 |
| 4 | Fire emergency plan | 1 | 0.1575 | 0.1575 |
| | Sum of A × B | (C) | | 1.0000 |
| | Relative weight of fire management | (D) | | 0.2500 |
| | Fire management scores | (C) × (D) | | 0.2500 |
| | Fire facility scores + fire management scores | | | 1.4320 |

**Table 8.** Fire probabilistic risk assessment of evacuation factor.

| Criteria | Evacuation Factor | Grade Assessment (A) | Weighting Attribute (B) | Scores Attribute (A × B) |
|---|---|---|---|---|
| Evacuation facility | | | | |
| 1 | Indoor fire exit stair | 5 | 0.1300 | 0.6500 |
| 2 | Fire exit door | 2 | 0.1040 | 0.2080 |
| 3 | Fire exit discharge | 1 | 0.0830 | 0.0830 |
| 4 | Emergency radio | 3 | 0.0390 | 0.1170 |
| 5 | Capacity of means of egress | 4 | 0.4540 | 1.8160 |
| 6 | Dead-end corridors | 5 | 0.1630 | 0.1630 |
| 7 | Escape equipment | 1 | 0.0270 | 0.0270 |
| | Sum of A × B | (C) | | 3.0640 |
| | Relative weight of evacuation facility | (D) | | 0.7310 |
| | Evacuation facility scores | (C) × (D) | | 2.2398 |
| Personal characteristics | | | | |
| 1 | Crowd density | 3 | 0.1400 | 0.4200 |
| 2 | Number of blind students per teacher | 5 | 0.2900 | 1.4500 |
| 3 | Degree of familiarity with building | 3 | 0.5700 | 1.7100 |
| | Sum of A × B | (C) | | 3.5800 |
| | Relative weight of personal characteristics | (D) | | 0.1880 |
| | Personal characteristics scores | (C) × (D) | | 0.6730 |
| Fire rescue | | | | |
| 1 | Distance from fire brigade | 5 | 0.5730 | 2.865 |
| 2 | Distance from hospital | 5 | 0.1400 | 0.700 |
| 3 | First aid kit | 5 | 0.2870 | 1.435 |
| | Sum of A × B | (C) | | 5.000 |
| | Relative weight of fire rescue | (D) | | 0.081 |
| | Fire rescue scores | (C) × (D) | | 0.4050 |
| | Evacuation facility + personal characteristics + fire rescue | | | 3.3178 |

The calculation of the consistency ratio was divided into the following steps.

(1)  Relative weight calculation and the maximum eigenvalue ($\lambda_{\max}$) for each matrix based on the number of decision criteria (*n*).

(2)  Consistency index calculation for each matrix based on the number of decision criteria (*n*) $CI = (\lambda_{\max} - n)/(n - 1) = (3.06 - 3)/(3 - 1) \approx 0.03$.

(3)  Consistency ratio calculation, which was $C.R. = CI/RI = 0.03/0.58 \approx 0.06$.

The fire control factor and evacuation factor were calculated using the same calculation method as the inherent fire hazard factor.

All fire safety factors from Expert Choice V.11 showed the value of consistency ratio (C.R.) to be less than 0.1, and the pairwise comparison matrix was assumed to have an acceptable consistency. The probabilistic risk of each fire safety factor was compared with the fire risk score in order to realize the risk status, as shown in Table 9.

**Table 9.** Fire probabilistic risk scores.

| Fire Probabilistic Risk Scores | 1 ≤ R ≤ 1.5 | 1.5 < R ≤ 2.5 | 2.5 < R ≤ 3.5 | 3.5 < R ≤ 4.5 | 4.5 < R ≤ 5 |
|---|---|---|---|---|---|
| Risk ranking | Class 1 | Class 2 | Class 3 | Class 4 | Class 5 |
| Risk status | worst | worse | good | better | best |

There were three main factors, which included inherent fire hazard, evacuation, and fire control factor. These factors were composed of several subfactors, and the subfactors were evaluated to provide a score and calculate weighting attribute. The AHP principle was used to calculate the relative weight of each factor. The results of score attribute summation and relative weight were multiplied to obtain the fire probabilistic risk scores. These fire probabilistic risk scores were classified regarding whether the risk scores are lower than 2.5, which is an unacceptable risk. The findings revealed that inherent fire hazard and evacuation factor were acceptable risks (good level of risk status), but the fire control factor was an unacceptable risk (worst level of risk status). The three main factors of fire hazard risk scores were 3.2830, 3.3178, and 1.4320, respectively, as shown in Figure 7. Therefore, the fire control factor was mitigated by the practical recommendations.

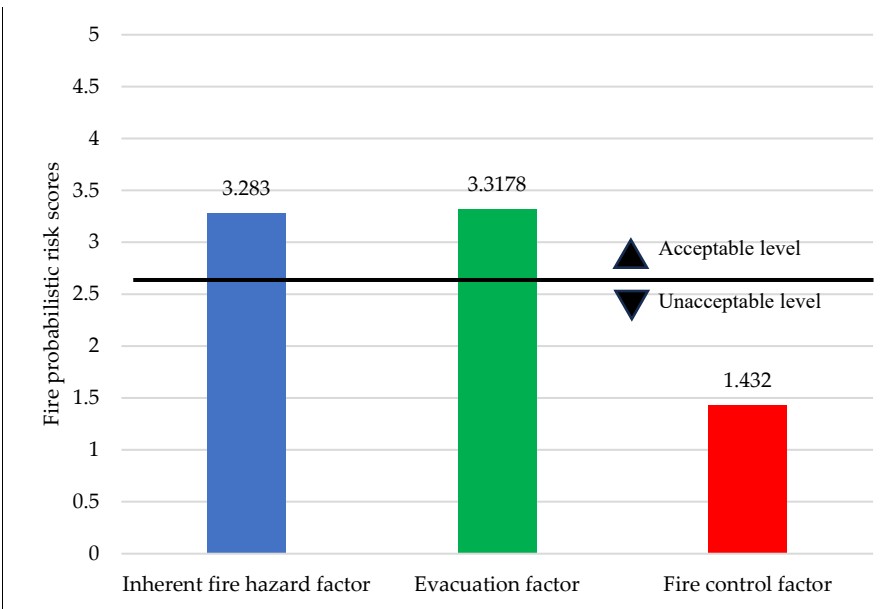

**Figure 7.** Fire probabilistic risk scores.

## 4. Discussion

Fires are the leading cause of death, serious injury, and property damage. Moreover, fire causes more damage among the blind, who have limited vision. Therefore, there is a need to assess the fire probabilistic risk in school for the blind buildings. Assessment of the scores was based on brainstorming from fire experts. Expert Choice V.11 implemented the AHP principle to compute the relative weight of each of the fire safety factors. Similarly, Ki-Chang Hyun et al. [24], Alberto Petruni et al. [25], Ali Kokangül, Ulviye Polat, and Cansu Dağsuyu [26], and Maria Garbuzova-Schlifter and Reinhard Madlener [27] all applied the AHP to calculate the relative weights for their decision goals. This study applied an AHP principle to determine the relative weight of each of the fire safety factors from the checklist and determine the probabilistic risk in a school for the blind building. This is the first application of the AHP principle to a fire checklist. No other research has been found to apply this, but most of them involved applications of the AHP principle to fault tree analysis (FTA) and event tree analysis (ETA) techniques. However, here, the reliability of the information was evaluated through extensive research and expert judgments to gain the relative weights of the fire safety factors to assess the fire probabilistic risk. As a result,

the probability of occurrence is even more clearly defined without having to make sense of probabilistic risk assessment.

In addition, Emre Akyuz, Ozcan Arslan, and Osman Turan [28] also applied the fuzzy logic to fault tree and event tree analysis of the risk for cargo liquefaction onboard ships. Their study is very similar to this one, but the technique used was simple for the general public to practice, making it easy to assess the fire probabilistic risk. The AHP principle is also suitable for application. Finally, Meghann Valeo et al. [29], Abbas Bahrami et al. [30], and Mohammad Hossein Memary Nashalji, Abbas Bahrami, and Habibollah Rahimi [31] also found that the checklist is a technique that can be applied for simple bridge security, occupational medicine status, and performance of industrial occupational health experts, respectively. This is similar to our study, which applied the checklist technique to assess the fire probabilistic risk. However, there are differences between these studies: the implementation of AHP was used to make decisions to determine the relative weight for reducing bias assessments (subjective probability), including obtaining the scores for assessment of the grade through brainstorming.

The fire probabilistic risk assessment revealed that of the inherent fire hazard factor (3.2830 out of 5), evacuation factor (3.3178 out of 5), and fire control factor (1.4320 out of 5), only the fire control factor was at an unacceptable level. The fire control factor consisted of two subfactors, which were fire facility and fire management. Fire facility (relative weight = 0.7500) had a greater relative weight than fire management (relative weight = 0.2500) based on calculations from Expert Choice V.11. When considering the details of fire facility, it was found that the scores for portable fire extinguisher and fire alarm and fire control linkage system were higher than other items. As for the details of fire management, it was discovered that the scores for fire safety inspection and basic fire-fighting were higher than other items. These findings indicate that both active and passive fire protection systems are still essential in ensuring fire safety, which is in accordance with Wenlong Li et al. [32], who stated that a fire risk assessment index system was suitable for the buildings under construction. The index weight was calculated in order to gain the fire risk assessment and set up measures for active and passive fire protection. There were also studies on fire risk assessment using the AHP principle. These assessments did not apply brainstorming from fire experts, which may affect the reliability of the fire risk assessment and fire prevention measures [33–35].

The practical suggestions for reducing the fire probabilistic risk in terms of the fire control factor included checking and testing all fire emergency protection and response equipment according to a monthly preventive maintenance plan, design and installation of portable fire extinguishers as per the NFPA10 [36] standard, making a suitable new floor plan including safety signs for the blind, and, most importantly, fire drills and basic firefighting must be carried out according to fire emergency plans in order for the blind to learn how to survive in the event of a fire.

## 5. Conclusions

Assessing the fire risk of control is important in the occupational health and safety field, and the results of fire probabilistic risk assessments must be reliable. Consequently, this paper focused on conducting an AHP using Expert Choice V.11 to determine the relative weights of fire safety factors in order to estimate fire probabilistic risk in a school for the blind building. As a result of the findings, especially for the fire control factor (fire facility and fire management), school authorities (school director, deputy director of the school, staff, etc.) must set up a risk management plan or standard operating procedures. The procedures should include potential failures of measures to prevent fire hazards as well as actions to be conducted to prevent injury and death in order to efficiently address the risks that are likely to arise from routine operational procedures. Both active and passive fire protection measures are still essential in ensuring fire safety after assessing the fire risk. Limitations of this study are based on the survey of only one school and brainstorming of fire experts for determining the grade assessment. Therefore, it may not represent the

actuality of the situation due to limited time and resources available. However, it represents the general facts.

In conclusion, this paper will help researchers and school administrators to decrease potential risks during study in the school. Further studies may be extended with quantitative risk assessment and fuzzy analytic hierarchy process (FAHP) approaches to manage uncertainty in a better way.

**Author Contributions:** Conceptualization, A.K. and R.P.; methodology, A.K.; software, R.P.; validation, A.K.; investigation, A.K.; data curation, R.P.; writing—original draft preparation, R.P.; writing—review and editing, A.K.; visualization, R.P.; supervision, A.K.; project administration, A.K. All authors have read and agreed to the published version of the manuscript.

**Funding:** This research received no external funding.

**Institutional Review Board Statement:** This study was reviewed and exempted by the Office of the Human Research Ethics Committee of Bangkok.

**Informed Consent Statement:** Not applicable.

**Data Availability Statement:** Data will be made available on request.

**Acknowledgments:** We thank the blind school that was the subject of this study, and it has not refrained from supporting the determination of assessing the fire probabilistic risk in our study. We also wish to thank the reviewers and editor in charge for their very constructive feedback.

**Conflicts of Interest:** The authors declare no conflict of interest.

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
