# Peer review of "Application of Analytic Hierarchy Process to Rank Fire Safety Factors for Assessing the Fire Probabilistic Risk in School for the Blind Building: A Case Study in Thailand"

_fire, doi:10.3390/fire6090354_

Round 1

Reviewer 1 Report (New Reviewer)

This paper conducts Analytic Hierarchy Process for determining the relative weight from fire safety factors to assess the fire probabilistic risk in school for the blind building. The inherent fire hazard factors and evacuation factors are acceptable risk while the fire control factors are unacceptable risk.

Here are some questions and comments that may improve the paper:

 1. This paper focuses on the Analytic Hierarchy Process application to calculate the weight of fire safety factors. However, the calculation results are mainly based on the software, and it is necessary to present some specific calculation process in detail.

 2. The result of fire probabilistic risk assessment is closely related to the suggestions for reducing the fire risk, which should be added some additional discussion in Section 3.

 3. The fire risk score in Table 3 and the description of on lines 199-203 do not match.

 4. Present your figs with larger fonts, especially Figs. 4-6.

 5. There are several typos and grammar mistakes, such as the sentence on line 195.

     6. The paper need be properly modified and rewritten.

It is noted that the manuscript needs careful editing by someone with expertise in technical English editing, particularly the English grammar and sentence structure.

Author Response

Dear Reviewer No.1,

First of all, we would like to thank you for the suggestion. Any your comments are very helpful. Please see the file attached the author's reply to previous reviewers.

Best regards,

Arroon Ketsakorn

Reviewer 2 Report (New Reviewer)

The manuscript sent for evaluation consists of 13 pages, including pages of text with 7 tables and 7 figures, and 1 pages for references (29). The main author does not have autocitations.

After reading the text, in general, I assess the scientific quality of the publication  well.

I suggest to add title of article: Case study in Thailand. 

The authors presented the results of theoretical research, specifically study was to assess the fire risk in school for the blinds.

Abstract would be to contain all the necessary information, such as methods, results, main statements. Please, add aim in abstract and add specifically results.

The visual documentation is illustrative and clear (except for Fig. 4, 5 and 6). 

Please, add aim of research. Please specify the novelty and aplication of your study in the introduction or conclusion.

Authors choose software for cooperative decision-making solution that is based on multi-criteria decision making from 1983. Why this software?  

The conclusions are brief. Please clearly specify the result obtained.

My question:

Line 247: What does mean “unacceptable level “Fire control factors”? These factors are described in Table 5 and Table 6. I think, that these factors are key important for fire safety and these factors may be applie for active and pasive prevention. Please, explain it.

In my opinion Figure 4, Figure5 and Figure 6 are not suitable for this form presentation. 

Author Response

Dear Reviewer No.2,

First of all, we would like to thank you for the suggestion. Any your comments are very helpful. Please see the file attached the author's reply to previous reviewers.

Best regards,

Arroon Ketsakorn

Reviewer 3 Report (New Reviewer)

Dear Authors,

In line with the proofreading criteria of the publisher, I prepared a reviewer’s report, which would be as follows:

The content of the proposed paper mostly meets the objectives of the journal.

Using the scientific methods (fire checklists, brainstorming and Analytic Hierarchy Process) applied in accordance with the author’s scientific objectives resulted useful scientific achievements.

The main strength of the study is that the authors – in the form of case study - systemically assess the fire risk in school for the blinds based on probabilistic risk assessment approaches.

The references used in the main chapters are relevant and assist the reader to understand the authors proposals. The illustrations used are regular and correct.

In addition to acknowledging the high-quality work, I recommend the following miner corrections:

1. Introduction. It is recommended to consider dividing paragraphs 30-94 into several parts for better clarity. At the end of Introduction, the main objectives of this study should be clearly and detailed presented, and the main conclusions of the article should be highlight. The number of references is should be increased.

3. Results. The values of the socially acceptable and unacceptable risk levels in Figure 7 require a more detailed explanation.

4. Discussion. This section is too brief and should be detailed. In this section, the experimental results should be clearly presented and detailed discussed. It is also recommended to consider dividing paragraphs 250-284 into several parts for better understanding.

5. Conclusions. It should be recommended to add some subsections, such as:

- contributions to theory,

- contributions to practitioners, like Department of Disaster Prevention and Mitigation, Ministry of Interior of Thailand, regional and local emergency services, school management),

- limitations and suggestions for future research.

Based on the above, after minor revision, I suggest the publication of reviewed article.

Minor editing of English language required.

Author Response

Dear Reviewer No.3,

First of all, we would like to thank you for the suggestion. Any your comments are very helpful. Please see the file attached the author's reply to previous reviewers.

Best regards,

Arroon Ketsakorn

Round 2

Reviewer 1 Report (New Reviewer)

The authors have addressed my concerns well, and the paper can be accepted in the current version.

This manuscript is a resubmission of an earlier submission. The following is a list of the peer review reports and author responses from that submission.

Round 1

Reviewer 1 Report

The study assesses the fire risk of a school for blind by using the methods of the fire checklists, brainstorming and AHP. The result shows that the risk of the fire control factors is unacceptable. This paper will help for decreasing potential fire risks of the school for blind. While I recommend a major revise for the paper considering the comments as follows.

1. In the part of Introduction, I think the literature review is not sufficient and detailed enough so that the readers are unable to aware the current situation of fire risk assessment.

2. Please illustrate the innovation of this paper because fire risk assessment has been extensively studies and the methods used in this study are not new.

3. In the Part of Results, the assessment of the scores were based on brainstorming from fire experts. I think the author should supplement the specific information of the experts, such as their professional experience and expertise to make the results credible. And how can the authors guarantee the reasonableness and accuracy of the experts’ score.

4. In the part of Discussions, the author discussed the methods used in the study. Why not discuss the countermeasures and suggestions for blind schools as a special place to improve the fire safety level based on the evaluation results.

Author Response

Point1: In the part of Introduction, I think the literature review is not sufficient and detailed enough so that the readers are unable to aware the current situation of fire risk assessment.

Response1: Please provide your response for Point 1 (in red).

Fires in Thailand tend to happen more frequently and severely mainly because of unsafe acts and unsafe conditions. For megacity with a high permanent and temporary population like Bangkok, fire management becomes more crucial. Fires may cause many injuries and casualties as well as damage to property and assets, let alone harm to the overall economic system. Department of Disaster Prevention and Mitigation, Ministry of Interior of Thailand [1], as the administrative body responsible for the city, undertakes the mission to protect population from public disasters and make life safer, report the number of fires between 2017 to 2022 amounted 13,593 cases which mainly occurred in the school, temple and government offices (40% of all buildings). Moreover, it causes more damage among the blinds who has a limited vision.

Previous studies in the fire probabilistic risk assessment sector have predicted the risk status in the building where has the occupants to able the fire evacuation when fire occurred. Both deterministic and stochastic fire risk assessment approach was used to determine the risk status without brainstorming and evaluating the consistency of fire expert’s judgement which causes the fire risk status may be inaccurate such as Prashant A/L Tharmarajan [2] studied on the essential aspects of fire safety management in high-rise buildings; this study used the checklist technique to survey the level of fire expert’s opinions for determination of fire safety management. However, the consistency of the opinions level was not assess to reduce the bias in the fire expert’s judgement, John A. Moore and M. Phil [3] conducted a study on the assessment of fire safety and evacuation assessment in nursing home; this paper used a building survey to establish fire safety facilities, analysis of the fire related documentation, interview with staff to establish existing fire safety procedures and fire risk assessment was carried out in this study. Just one aspect that this study has not addressed: the acquisition of fire expert’s opinion levels is undertaken to reduce bias in the assessment of individual fire expert ratings, Yeung Cho Hung [4] was to study on fire safety management of public rental housing in Hong Kong. Fault Tree Analysis (FTA) was used to study the effect of fire safety management on the reduction in fire risk level. FTA is a method of identifying the possible cause of a system failure. By using FTA, assessing team can determine what factors contributed to an event (known as a failure or top event), and the probability of it occurring. However, the various probabilities are estimated to evaluate the risk level without considering the consistency assessment to reduce the bias individual fire expert’s judgement, Tanima Abdul Wahed [5] was to focus on impact of facility management on fire safety crisis in industry. A qualitative questionnaire survey was conducted to achieve the real picture of fire management practices in Bangladesh. But there are no reference case studies demonstrating an application of the fire probabilistic risk assessment. There is also a study by Ayyappa Thejus Mohan [6] focused on risk acceptance in fire safety engineering. Traditional deterministic fire engineering was used to be a tool for data collection from experience of the fire safety profession as well as obtaining through a continuous process of trial and error for assessing the fire risks. However, likelihood of harm for assessing the risk has not assessed the consistency in individual scoring. All above research focused on the fire probabilistic risk assessment and management in different buildings but there is no focus on a new tool used for calculating the level of risk. It obviously can be seen that there is still a lack of reliable fire safety assessment tool for reduction of bias individual fire expert’s judgement. This is an important thing to ensure the safety of occupants in the building especially blind people who have limited vision. Assessing the fire risk is important in occupational health and safety. Therefore, this study aimed to assess fire risk in school for the blind building including applying fire probabilistic risk assessment technique using the checklist method, which is a hazard identification technique used to reduce failure by compensating for potential limits of human memory and attention. It helps to ensure consistency and completeness in carrying out a task. This technique examines fire hazards according to laws and standards and therefore appropriate and consistent with the task of applying it for risk assessment as compared to other techniques. However, the checklist has limitations in determining the probability of the event, which sometimes some fire safety factors are unable to determine these values. Therefore, it is necessary to apply other techniques to determine the probability of events.

The Analytic Hierarchy Process (AHP) proposed by Thomas Saaty [7] is a versatile tool for dealing with complex decision-making problems. It is a hierarchical structure of goals, objectives, main criteria, sub-criteria, and alternative. The AHP assists deciders look for one the best matches for their goal by eliminating biased decision. It provides an extensive and methodical structure for construction a decision issue, for illustrating and estimating its elements, for implicating those elements to overall goals, and for assessing alternative solutions. The integration of AHP and expert analysis is frequently found in several studies, especially in the fields of waste management [8-9], environmental deterioration [10], project management selection [11-12], and disaster response [13], while the applications in the domain of determination of fire safety factors for assessing the fire probabilistic risk are less consolidated. This study eventually has applied an AHP to determine the relative weight of each fire hazard factors from checklist and to determine the probabilistic risk of fire. It also proposed measures for managing the fire risk in school for the blind building. One of the advantages of AHP application is its stability and flexibility regarding changes within and addition to the hierarchy of structure. Another advantage of AHP is its ability to rank main factors in accordance with the needs of policy-makers leading to more precise decision-making. However, AHP may have some weak points. One of them is the complexity of the method, which may be inconvenient in implementation. When more than one person is working with this method, it can be complicated by different opinions on weighting each of the criteria. 

Point2: Please illustrate the innovation of this paper because fire risk assessment has been extensively studies and the methos used in this study are not new.

Response2: Please provide your response for Point 2 (in red).

Previous studies in the fire probabilistic risk assessment sector have predicted the risk status in the building where has the occupants to able the fire evacuation when fire occurred. Both deterministic and stochastic fire risk assessment approach was used to determine the risk status without brainstorming and evaluating the consistency of fire expert’s judgement which causes the fire risk status may be inaccurate.  Therefore, this study aimed to assess fire risk in school for the blind building including applying fire probabilistic risk assessment technique using the checklist method, which is a hazard identification technique used to reduce failure by compensating for potential limits of human memory and attention. It helps to ensure consistency and completeness in carrying out a task. This technique examines fire hazards according to laws and standards and therefore appropriate and consistent with the task of applying it for risk assessment as compared to other techniques. 

Point3: In the part of Results, the assessment of the scores were based on brainstorming from fire experts. I think the author should supplement the specific information of the experts, such as their professional experience and expertise to make the results credible. And how can the authors guarantee the reasonableness and accuracy of the experts' score. 

Response3: Please provide your response for Point 3 (in red).

Assessment the scores were based on brainstorming from fire safety experts. The fire safety experts consisted of fire safety and building services (2 persons), fire fighters (2 persons), fire safety consultants (2 persons), and architect (1 person). All experts were experienced in fire safety more than 5 years. 

All fire safety factors from Expert Choice V.11 showed the value of Consistency Ratio (C.R.) less than 0.1, the pairwise comparison matrix was assumed to have an acceptable consistency. 

Point4: In the part of Discussions, the author discussed the methods used in the study. Why not discuss the countermeasures and suggestions for blind school as a special place to improve the fire safety level based on the evaluation results.

Response4: Please provide your response for Point 4 (in red).

The suggestions for reducing the fire risk included check and test all fire emergency protection and response equipment according to monthly preventive maintenance plan, design and installation of portable fire extinguisher as NFPA10 [29] standard, make a suitable new floor plan including safety sign for the blinds, and most importantly fire drills and basic firefighting must be carried out according to fire emergency plans in order for the blinds to learn how to survive in the event of a fire. 

Reviewer 2 Report

Detailed remarks are listed below:

15 – please consider changes in describing disabilities – like people with disability and so on…

18 – the risk evaluation method might be unknow for readers. Maybe it is not necessary to give specific numbers without explanation in abstract

72 – the description of the building is not sufficient. There are no information about the safety factors. Are the staircases enclosed?

83 – will be the checklist published as an appendix?

176 – the results are totally unclear.  Where they come from? There are no fire safety details. Only some numbers – which origin is not described.

242 – I can’t find any conclusions on fire safety for people with sight impairment. Nor for the buildings they are staying in…

Overall the proposed paper shows a process of application of a method of risk evaluation. The description scraps the surface of the problem. It gives no insights. And is very general

Please look for proper description of people with different disabilities

Author Response

Point1: Please consider changes in describing disabilities – like people with disability and so on…

Response1: Please consider the explanation for Point 1 (in red).

We have already changed the disability to the blinds who has a limited vision. 

Point2: the risk evaluation method might be unknow for readers. Maybe it is not necessary to give specific numbers without explanation in abstract.

Response2: Please consider the explanation for Point 2 (in red).

The findings revealed an inherent fire hazard factors and evacuation factors were acceptable risk except the fire control factors was unacceptable risk. The unacceptable risk may cause an impact on life, health, property and public communities. 

Point3:  the description of the building is not sufficient. There are no information about the safety factors. Are the staircases enclosed?

Response3: Please consider the explanation for Point 3 (in red).

The study area was a classroom of a blind school building according to the Building Act, B.E. 2522 of Thailand. This classroom is ventilated air natural. There are three stories layer, 79.5 m. long, 16 m. wide and 3 m. high, the 1st and 3rd floor has three fire escape stair exits, the 2nd floor has two fire escape stair exits, the fire escape stair exit was blocked off one fire escape stair exit to create a new classroom on the 2nd floor as shown in Figure 1. 

Point4: will be the checklist published as an appendix?

Response4: Please consider the explanation for Point 4.

The items of checklist were shown in Table 4-6 already.

Point5: the results are totally unclear.  Where they come from? There are no fire safety details. Only some numbers – which origin is not described.

Response5: Please consider the explanation for Point 5 (in red).

The fire risk score was considered in two elements. Namely, the assessment grade scores were based on brainstorming from seven fire safety experts and with meaning that the probability level was remote possibility (Score=5), possible but not possibility (Score=4), moderately possibility (Score=3), important possibility (Score=2), and most possibility (Score=1), respectively. The scores of fire probabilistic risk assessment were shown in Table 3 and relative weights of fire probabilistic risk assessment indicator were obtained from the calculation by Expert Choice V.11. The determination of each weighting criteria and weighting attribute were assumed a pairwise comparison matrix provided by decision-makers based on the Delphi MAH technique developed by the Rand Corporation Company and the Maximize Agreement Heuristic (MAH). This technique was consisted of three steps; the acquisition the numerical data from the fire safety experts, all decision data were averaged, and average values from all the decision-makers were considered.  The probabilistic risk assessment was performed in Equation (5) and unacceptable risk level (score less than 2.5) requires further mitigation measures.

Point6: I can’t find any conclusions on fire safety for people with sight impairment. Nor for the buildings they are staying in…

Response6: Please consider the explanation for Point 6 (in red).

As a result of the findings especially fire control factors (fire facility and fire management), school authorities (school director, deputy director of the school, staff, etc.) must set up a risk management plan or standard operating procedures. The procedures should include potential failures of measures to prevent fire hazards and actions to be conducted to prevent injury and death in order to efficiently address the risks that are likely to arise from routine operational procedures. 

The suggestions for reducing the fire risk included check and test all fire emergency protection and response equipment according to monthly preventive maintenance plan, design and installation of portable fire extinguisher as NFPA10 [29] standard, make a suitable new floor plan including safety sign for the blinds, and most importantly fire drills and basic firefighting must be carried out according to fire emergency plans in order for the blinds to learn how to survive in the event of a fire. 

Reviewer 3 Report

This paper analyzes the probability of fire in a blind school building, the risk assessment of fire, and the importance ranking of the factors affecting fire safety through hierarchical analysis. A school in Bangkok, Thailand, was selected in the field and the materials and design of the school building were explored to analyze and assess the factors affecting evacuation after a fire, which is useful for understanding the fire spread and evacuation process.  Below are some comments:

1. In the introduction section of this paper, the introduction to the literature on the assessment of fires lacks the introduction of the superiority of AHP over other assessment methods.

2. The building data in the presentation of the school plan in Figure 1 is not clear.

3. There is a formatting problem with the serial numbers of the different pictures in Figure 1, some are bolded, and some are not, please standardize the format.

4. After the article talks about the weights of different factors regarding fire hazards, can reasonable suggestions be given based on the assessment results.

Author Response

Point1: in the introduction section of this paper, the introduction to the literature on the assessment of fires lacks the introduction of the superiority of AHP over other assessment methods.

Response1: Please consider the explanation for Point1 (in red).

The Analytic Hierarchy Process (AHP) proposed by Thomas Saaty [7] is a versatile tool for dealing with complex decision-making problems. It is a hierarchical structure of goals, objectives, main criteria, sub-criteria, and alternative. The AHP assists deciders look for one the best matches for their goal by eliminating biased decision. It provides an extensive and methodical structure for construction a decision issue, for illustrating and estimating its elements, for implicating those elements to overall goals, and for assessing alternative solutions. The integration of AHP and expert analysis is frequently found in several studies, especially in the fields of waste management [8-9], environmental deterioration [10], project management selection [11-12], and disaster response [13], while the applications in the domain of determination of fire safety factors for assessing the fire probabilistic risk are less consolidated. This study eventually has applied an AHP to determine the relative weight of each fire hazard factors from checklist and to determine the probabilistic risk of fire. It also proposed measures for managing the fire risk in school for the blind building. One of the advantages of AHP application is its stability and flexibility regarding changes within and addition to the hierarchy of structure. Another advantage of AHP is its ability to rank main factors in accordance with the needs of policy-makers leading to more precise decision-making. However, AHP may have some weak points. One of them is the complexity of the method, which may be inconvenient in implementation. When more than one person is working with this method, it can be complicated by different opinions on weighting each of the criteria.

Point2: The building data in the presentation of the school plan in Figure 1 is not clear.

Response2: Please consider the explanation for Point2.

We have already modified the school plan as shown file attachment.

Point3: There is a formatting problem with the serial numbers of the different pictures in Figure 1, some are bolded, and some are not, please standardize the format.

Response3: Please consider the explanation for Point3.

We have already revised the format as shown file attachment.

Point4: After the article talks about the weights of different factors regarding fire hazards, can reasonable suggestions be given based on the assessment results.

Response4: Please consider the explanation for Point4 (in red).

The suggestions for reducing the fire risk included check and test all fire emergency protection and response equipment according to monthly preventive maintenance plan, design and installation of portable fire extinguisher as NFPA10 [29] standard, make a suitable new floor plan including safety sign for the blinds, and most importantly fire drills and basic firefighting must be carried out according to fire emergency plans in order for the blinds to learn how to survive in the event of a fire. 

Round 2

Reviewer 1 Report

Accept.

Reviewer 2 Report

Second review:

2 – The title. Please consider using phrase “school building for the blind”

112 – the drawings are illegible

Thank you for the provided changes. Unfortunately, the proposed paper doesn’t show much improvement. If it was focused only on expert risk evaluation method t0 than it could be considered as valuable one. But if it tries to give conclusions on safety factors based on unknown responses from chosen exerts – than it fails. When reading the discussion chapter there are described suggestion for some general and obvious improvements which can’t be find anywhere else in the paper. It is unclear what should be done to increase the safety factor and what value of this factor would be achieved if those propositions would be incorporated.

Please consider focusing on the methodology only or elaborating real conclusions for safety improvement with additional evaluation of its cost and importance.

And finally, please change the way you describe people with visual impairments.

Please change the way you describe people with visual impairments.